

# Optimal cultivation measures for maize production in the drylands of the Loess Plateau

Wenbo Mi, Gang Zhao, Jianjun Zhang, Yi Dang, Lei Wang, Gang Zhou, Shuying Wang, Shangzhong Li, Tinglu Fan, Jingyu Hu and Xujiao Zhou

Institute of Dryland, Gansu Academy of Agricultural Sciences, Lanzhou, China

## ABSTRACT

Proper nutrient management and planting density can effectively improve crop yields and economic efficiency. Optimized fertilization and high-density planting measures have gained attention for their ability to balance high crop yields with efficient nitrogen utilization. High and stable yield of maize is an important guarantee for global food security, and fertilizer and planting density are important factors affecting yield components. In this study, dryland maize (*Zea mays* L.) in the Loess Plateau was used as the research material, and a three-factor, five-level quadratic orthogonal rotated combination design with organic fertilizer, nitrogen fertilizer and density were set up for a field trial. Regression models of organic fertilizer, nitrogen fertilizer, density and maize yield were established. The results showed that the contribution of each factor to maize yield was in the order of density > nitrogen fertilizer > organic fertilizer. The contribution rates are as follows: 90.02, 89.23, and 79.66, respectively. The univariate analysis concluded that maize yield increased with increasing application of organic fertilizer. Nitrogen fertilizer and density positively affected maize yield within a certain range. Maximum yield is achieved when nitrogen fertilizer and planting density are set at levels 150 kg hm$^{-2}$ and 82,500 plants hm$^{-2}$, respectively. The analysis of the intercropping effects showed that organic fertilizer-nitrogen fertilizer, organic fertilizer-density, and nitrogen fertilizer-density intercropping had a synergistic effect on maize yield. Optimisation analysis of different cultivation measures was used to obtain the optimum range of treatments for yields higher than 15,000 kg hm$^{-2}$: 6,429.30–7,895.10 kg hm$^{-2}$ for organic fertilizer application, 159.30–203.55 kg hm$^{-2}$ for nitrogen fertilizer application, and 72,465–80,940 plants hm$^{-2}$ for planting density. This study explored the effects of different ratios and interactions of organic fertilizer, nitrogen fertilizer and density on maize yields, and screened out the scientific level of dense fertilizer to provide theoretical basis and practical experience for the standardised cultivation of maize in the Loess Plateau.

# INTRODUCTION

As the world's second largest maize (*Zea mays* L.) producer, China occupies an important position in the global maize industry in terms of planting scale, output, and consumption

Corresponding author
Yi Dang, 22126670@qq.com

demand, which has a profound impact on the balance of supply and demand in the international market and food security (*Ai et al., 2024*). The Loess Plateau in north-west China is typical of rain-fed agriculture, accounting for about 40% of the country's dryland area, and is one of China's important food-producing areas (*Xue et al., 2017*). Maize, as one of the world's major cereals, is also the most common food crop in the Loess Plateau (*Guo et al., 2021*). Maize cultivation not only ensures food security supply in the Loess Plateau, but also affects and restricts water allocation and ecological environment construction in the region to a certain extent (*Jun et al., 2007*). However, factors such as the region's particular geography, soil type and climatic conditions have created limitations in maize production (*Qiang et al., 2019*; *Zheng et al., 2018*). Therefore, a comprehensive understanding of the constraints of maize production in the Loess Plateau and the adoption of appropriate cultivation measures to improve maize yield are of great significance to food production in the Loess Plateau.

Poor soil fertility is one of the key factors limiting crop productivity in the Loess Plateau (*Wang, Yan & Gu, 2018*; *Han et al., 2010*). Fertilizer application meets the nutrient requirements of the crop during growth and development and plays an important role in determining maize yield (*Haque et al., 2024*; *Montemurro et al., 2007*; *Pakurar et al., 2004*). Nitrogen fertilizer, as the most applied fertilizer, is highly effective in increasing crop yields. In the continuing quest for higher yields, nitrogen fertilizer inputs are also increasing (*Takács et al., 2007*; *Wang et al., 2023*). However, excessive nitrogen fertilizer application not only fails to effectively increase maize yield, but also causes a decline in nitrogen fertilizer use efficiency and soil nutrient imbalance (*Haque & Hoque, 2023*). It has also led to increasing secondary salinisation of soil and ecological pollution (*Man-Man, 2011*; *Min et al., 2014*). To solve this problem, the application of organic fertilizer has received widespread attention.

Organic fertilizers contain a variety of active substances that can dissolve insoluble compounds in the soil and have the advantages of increasing soil fertility, improving the physical and chemical properties of the soil, maintaining the balance of soil nutrients and reducing environmental pollution (*Aluoch et al., 2022*; *Haque et al., 2018*; *Xiao et al., 2021*). Research has shown that organic fertilizers with nitrogen have a significant effect in improving soil nutrients and increasing maize yields (*Ming et al., 2023*). However, the amount of organic fertilizer and nitrogen fertilizer applied varies according to the growing area. In northeast China, where soil fertility is adequate and organic matter content is high, the application of organic fertilizers has limited yield enhancement. Nitrogen fertilizer application at 180 kg hm$^{-2}$ significantly increased maize yield. In northwest China, the application of organic fertilizers significantly improves soil structure and enhances water and fertilizer retention. A mixture of N and organic fertilizers avoids rapid nutrient loss. The optimum N application for maize yield in this region has been as high as 270 kg hm$^{-2}$ (*Shen et al., 2005*; *Wei et al., 2004*). However, the choice of fertilizer application rates to improve maize yields in the Loess Plateau is an urgent issue that needs to be addressed.

Planting density is also one of the important cultivation measures that affect crop yields. Too low a planting density not only can not effectively use water, fertilizer, light and heat, but also too large a gap between plants will make rainwater erosion of the soil, leading to

soil erosion (*Barker & Edwards, 2010*). Excessive planting densities are not only ineffective in increasing crop yields, but also destabilise maize production by disrupting the soil water balance in dryland ecosystems (*Guo et al., 2021*; *Li et al., 2023*; *Sárvári, Hallof & Molnár, 2007*; *Salmerón et al., 2012*). It also increases the risk of plant diseases and pests. Therefore, screening for a reasonable planting density is crucial for improving maize yield in the Loess Plateau.

Currently, research on maize yield improvement in the Loess Plateau primarily focuses on single-factor studies (such as optimized nitrogen management, plastic film mulching innovations, and density-adapted cultivation) or dual-factor approaches (like organic–inorganic fertilizer combinations and density-cultivar matching). However, there have been few reports investigating the interactive effects of organic fertilizer, nitrogen application, and planting density on maize yield (*Wang et al., 2018*; *Han et al., 2018*). In this study, dry-crop maize on the Loess Plateau was taken as the research object, and the design of orthogonal rotating combinations through quadratic regression at five levels for three factors of organic fertilizer, nitrogen fertilizer and density were carried out. The effects of fertilizer application and planting density on maize yield and the interaction effects were investigated. In order to propose the optimal agronomic measures for high-yield cultivation of maize, and to provide theoretical basis for efficient production of dryland maize in the Loess Plateau.

## MATERIALS AND METHODS

### Experimental sites
The experiment was conducted from April 2022 to October 2024 at the National Soil Quality Observation Experimental Station (35°30′N, 107°29′E) of the Dryland Farming Institute, Gansu Academy of Agricultural Sciences. The area is located at an altitude of 1,279 m, with the soil being black loessial soil. The organic matter and total nitrogen content are 10.7 g kg$^{-1}$ and 0.91 g kg$^{-1}$, respectively, while the available nitrogen, available phosphorus, and available potassium contents are 91.3 mg kg$^{-1}$, 11.8 mg kg$^{-1}$, and 228.3 mg kg$^{-1}$, respectively, indicating medium fertility. The average annual precipitation is 533.4 mm, with 60% of the rainfall occurring between July and September, making it a typical rain-fed dryland farming area. The monthly rainfall distribution across different years is shown in Fig. 1.

### Experimental materials
The experimental material is the maize variety *Xianyu 1483*, with a 100-kernel weight of 37.55 g, a fertility period of 128 days, and a planting density of 4,500 plants hm$^{-2}$. It was provided by the Institute of Dryland Agriculture, Gansu Academy of Agricultural Sciences.

### Experimental design
The experimental treatments consisted of three factors at five levels (coded as −1.682, −1, 0, 1, 1.682). The corresponding levels for each factor were as follows: Organic fertilizer: M4: 2,220, M1: 3,570, M2: 6,000, M3: 8,250, M5: 9,780 kg hm$^{-2}$. Nitrogen fertilizer: N4: 23.85, N1: 75, N2: 150, N3: 225, N5: 276.15 kg hm$^{-2}$. Planting density:
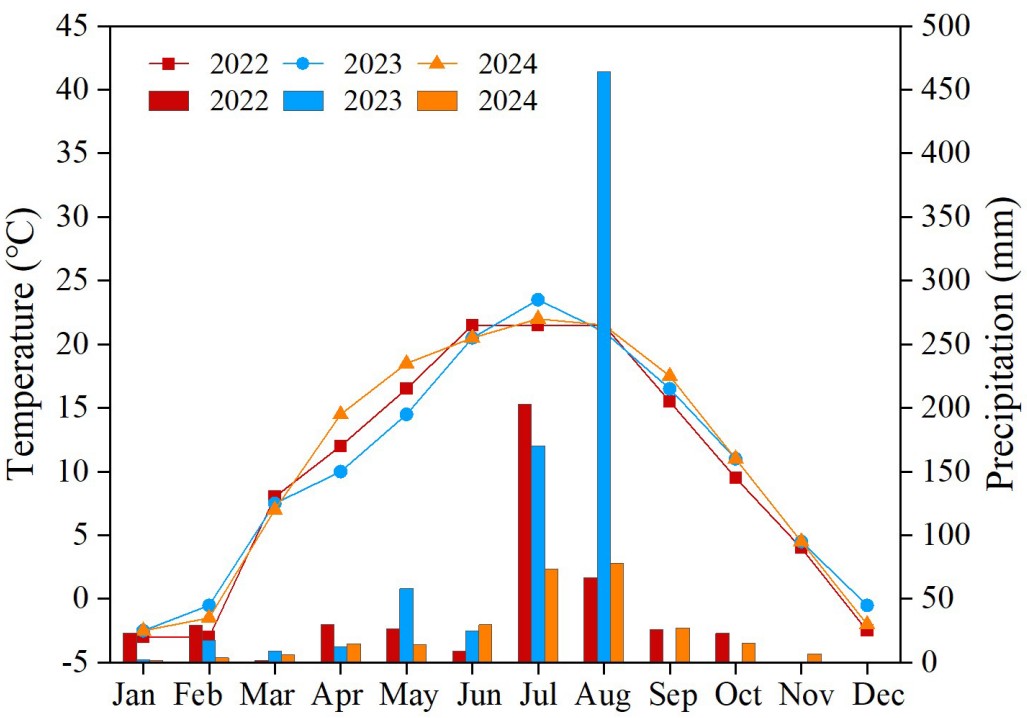

**Figure 1** Monthly rainfall and temperature by year.

**Table 1** Factor level table.

| Factor | Level code ($\gamma = 1.682$) | | | | |
|---|---|---|---|---|---|
| | $-\gamma$ | $-1$ | $0$ | $1$ | $+\gamma$ |
| Manure (M) | 2,220 | 3,750 | 6,000 | 8,250 | 9,780 |
| Nitrogen (N) | 23.85 | 75 | 150 | 225 | 276.15 |
| Density (D) | 42,270 | 52,500 | 67,500 | 82,500 | 92,730 |

D4: 42,270, D1: 52,500, D2: 67,500, D3: 82,500, D5: 92,730 plants hm$^{-2}$. According to the rotatable orthogonal experimental design, which is an experimental design method used to study systems with multiple factors and multiple levels. It selects a subset of representative experimental points from the full factorial experiments based on the principle of orthogonality. These representative points are characterized by uniform distribution across the experimental domain and balanced comparability, ensuring both efficiency and statistical validity in analyzing factor effects. A total of 23 treatments were arranged. For operational convenience, treatments 15–23 were combined, resulting in 15 plots. Each plot measured 20 m × 4.8 m = 96 m$^2$, with a spacing of 0.5 m between plots. All treatments were conducted under wide plastic film mulching, with a film width of 1.2 m and four mulched planting strips per plot. The phosphorus application rate was 120 kg hm$^{-2}$ (P$_2$O$_5$). The coded levels of the experimental factors are presented in Table 1, while the specific experimental design and implementation plan are detailed in Table 2.

**Table 2  Experimental design and implementation plan.**

| Plot No. | Text design | | | Applied rate | | |
|---|---|---|---|---|---|---|
| | M | N | D | M (kg hm$^{-2}$) | N (kg hm$^{-2}$) | D (plants hm$^{-2}$) |
| 1 | −1 | −1 | −1 | 3,750 | 75 | 52,500 |
| 2 | 1 | −1 | −1 | 8,250 | 75 | 52,500 |
| 3 | −1 | 1 | −1 | 3,750 | 225 | 52,500 |
| 4 | 1 | 1 | −1 | 8,250 | 225 | 52,500 |
| 5 | −1 | −1 | 1 | 3,750 | 75 | 82,500 |
| 6 | 1 | −1 | 1 | 8,250 | 75 | 82,500 |
| 7 | −1 | 1 | 1 | 3,750 | 225 | 82,500 |
| 8 | 1 | 1 | 1 | 8,250 | 225 | 82,500 |
| 9 | −1.682 | 0 | 0 | 2,220 | 150 | 67,500 |
| 10 | 1.682 | 0 | 0 | 9,780 | 150 | 67,500 |
| 11 | 0 | −1.682 | 0 | 6,000 | 23.85 | 67,500 |
| 12 | 0 | 1.682 | 0 | 6,000 | 276.15 | 67,500 |
| 13 | 0 | 0 | −1.682 | 6,000 | 150 | 42,270 |
| 14 | 0 | 0 | 1.682 | 6,000 | 150 | 92,730 |
| 15 | 0 | 0 | 0 | 6,000 | 150 | 67,500 |
| 16 | 0 | 0 | 0 | 6,000 | 150 | 67,500 |
| 17 | 0 | 0 | 0 | 6,000 | 150 | 67,500 |
| 18 | 0 | 0 | 0 | 6,000 | 150 | 67,500 |
| 19 | 0 | 0 | 0 | 6,000 | 150 | 67,500 |
| 20 | 0 | 0 | 0 | 6,000 | 150 | 67,500 |
| 21 | 0 | 0 | 0 | 6,000 | 150 | 67,500 |
| 22 | 0 | 0 | 0 | 6,000 | 150 | 67,500 |
| 23 | 0 | 0 | 0 | 6,000 | 150 | 67,500 |

## Experimental methods

Field preparation and plastic film mulching were carried out in early April during 2022–2024. Seeds were sown using a dibbler in mid-April, with harvesting conducted in late September. For yield assessment, a representative sampling area with uniform plant growth was selected from each plot. Within this area, 30 plants were systematically sampled row by row, with three replicates per plot. After threshing, grain moisture content was determined using a grain moisture meter (PM-8188-A), and the yield per unit area was calculated at a standardized moisture content of 14%.

## Statistics analysis

The experimental data were organized, statistically analyzed, and subjected to variance analysis using Microsoft Excel 2010 (Microsoft, Redmond, WA, USA) and SPSS 27.0 (IBM Corp, Armonk, NY, USA) statistical analysis software. Graphs were created using Origin 2021 software.

**Table 3  Yield of different years under different treatments.**

| Plot No. | Treatments | Yield (kg hm$^{-2}$) | | |
|---|---|---|---|---|
| | | 2022 | 2023 | 2024 |
| 1 | M1N1D1 | 11,116.05 | 11,923.58 | 10,943.25 |
| 2 | M3N1D1 | 11,642.27 | 12,018.45 | 11,029.67 |
| 3 | M1N3D1 | 12,258.56 | 12,247.65 | 11,287.11 |
| 4 | M3N3D1 | 12,446.73 | 12,843.45 | 12,367.82 |
| 5 | M1N1D3 | 14,178.51 | 14,662.36 | 13,368.45 |
| 6 | M3N1D3 | 15,718.46 | 15,549.26 | 13,506.43 |
| 7 | M1N3D3 | 16,240.50 | 15,814.80 | 14,764.65 |
| 8 | M3N3D3 | 16,940.33 | 16,115.25 | 15,345.01 |
| 9 | M4N2D2 | 14,699.40 | 14,352.11 | 13,898.62 |
| 10 | M5N2D2 | 16,602.21 | 15,630.12 | 14,798.70 |
| 11 | M2N4D2 | 13,969.50 | 13,782.90 | 12,918.15 |
| 12 | M2N5D2 | 14,984.10 | 14,313.60 | 13,882.73 |
| 13 | M2N2D4 | 10,324.76 | 9,907.35 | 9,925.28 |
| 14 | M2N2D5 | 14,525.10 | 14,992.68 | 14,376.15 |
| 15 | M2N2D2 | 15,228.60 | 15,173.41 | 14,928.00 |
| 16 | M2N2D2 | 15,068.10 | 15,363.15 | 14,724.45 |
| 17 | M2N2D2 | 15,078.45 | 15,214.65 | 14,676.00 |
| 18 | M2N2D2 | 14,876.85 | 14,994.75 | 14,746.05 |
| 19 | M2N2D2 | 15,285.45 | 15,414.15 | 14,886.75 |
| 20 | M2N2D2 | 14,892.75 | 14,840.70 | 15,004.80 |
| 21 | M2N2D2 | 14,908.35 | 14,957.70 | 14,900.40 |
| 22 | M2N2D2 | 15,290.25 | 15,272.10 | 14,825.55 |
| 23 | M2N2D2 | 15,349.80 | 15,361.65 | 14,840.40 |

# RESULTS

## Modelling and testing

Based on the experimental results (Table 3), a regression model of yield (Y) with organic fertilizer ($X_1$), nitrogen fertilizer ($X_2$) and density ($X_3$) was established.

$Y_{2022} = 15,115.402 + 450.621X_1 + 507.927X_2 + 1,660.474X_3 + 127.222X_1^2 - 287.767X_2^2 - 1,013.066X_3^2 - 147.271X_1X_2 + 109.674X_1X_3 + 167.111X_2X_3$

$Y_{2023} = 15,173.730 + 294.886X_1 + 275.302X_2 + 1,586.006X_3 - 34.898X_1^2 - 368.183X_2^2 - 933.130X_3^2 - 10.690X_1X_2 + 62.085X_1X_3 + 71.170X_2X_3$

$Y_{2024} = 14,845.102 + 248.890X_1 + 478.774X_2 + 1,379.610X_3 - 251.469X_1^2 - 586.648X_2^2 - 1,028.403X_3^2 - 179.584X_1X_2 - 56.099X_1X_3 + 194.096X_2X_3$.

The regression coefficients, total regression coefficients and lack of fit of the equations were tested separately by $F$-test to obtain the ANOVA table (Table 4). The ANOVA results showed that the total regression coefficients of equations $Y_{2022}$, $Y_{2023}$ and $Y_{2024}$ reached the highly significant level ($P < 0.001$), and the lack of fit did not reach the significant level ($P > 0.05$), which indicated that the models of equations Y2022, Y2023 and Y2024 were valid, and the regression equations were reasonably and reliably fitted. Finally, the optimised

**Table 4  Analysis of variance between yield and various treatments ($F$-value).**

| Treatments | 2022 | 2023 | 2024 |
|---|---|---|---|
| $X_1$ | 32.233[***] | 10.623[**] | 36.754[***] |
| $X_2$ | 43.730[***] | 26.299[***] | 203.515[***] |
| $X_3$ | 486.857[***] | 376.536[***] | 1,121.828[***] |
| $X_1^2$ | 1.093 | 0.559 | 5.649[*] |
| $X_2^2$ | 5.473[*] | 5.900[*] | 8.513[*] |
| $X_3^2$ | 8.707[**] | 14.592[***] | 14.923[**] |
| $X_1X_2$ | 4.961 | 0.748 | 22.950[***] |
| $X_1X_3$ | 8.316[*] | 0.983 | 2.240 |
| $X_2X_3$ | 6.388[*] | 3.584 | 26.809[***] |
| R ($F_2$) | 32.274[***] | 94.372[***] | 78.933[***] |
| Lack of Fit 1f ($F_1$) | 0.901 | 3.584 | 3.387 |

Notes.
[*]$0.01 < P < 0.05$.
[**]$0.001 < P < 0.01$.
[***]$P < 0.001$.

regression equations were obtained after the ANOVA test to eliminate insignificant terms ($P > 0.05$).

$$Y'_{2022} = 15{,}115.402+450.621X_1+507.927X_2+1{,}660.474X_3-287.767X_2^2-1{,}013.066X_3^2+109.674X_1X_3+167.111X_2X_3$$

$$Y'_{2023} = 15{,}173.730+294.886X_1+275.302X_2+1{,}586.006X_3-368.183X_2^2-933.130X_3^2$$

$$Y'_{2024} = 14{,}845.102+248.890X_1+478.774X_2+1{,}379.610X_3-251.469X_1^2-586.648X_2^2-1{,}028.403X_3^2-179.584X_1X_2+194.096X_2X_3.$$

## Factor main effects analysis

The contribution rate of each factor to yield was calculated based on the $F$ value of each treatment, and the effect of each factor on yield in each year could be obtained by calculating the contribution rate of each factor to maize yield in different years (Table 5). The effects of organic fertilizer, nitrogen fertilizer, and density on maize yield exhibited significant interannual variations. Quantitative analysis of the 2022–2024 growing seasons revealed a consistent hierarchy of contributing factors: density > nitrogen fertilizer > organic fertilizer. Meanwhile, the contribution of density to maize yield was much greater than that of fertilizer.

## One-way effects analysis

Dimensionality reduction method was used to study the effect of organic fertilizer, nitrogen fertilizer and density on maize yield, fixing the level of other factors at zero, the effect of a single factor on maize yield could be obtained. Fixing the nitrogen fertilizer and density factors at zero level gives an equation for the effect of organic fertilizer on maize yield:

$$Y_{2022} = 15{,}115.402+450.621X_1$$
$$Y_{2023} = 15{,}173.730+294.886X_1$$

**Table 5  Single factor contribution rate (%).**

| Treatments | 2022 | 2023 | 2024 |
|---|---|---|---|
| $X_1$ | 4.93 | 4.51 | 5.79 |
| $X_2$ | 5.05 | 6.26 | 14.54 |
| $X_3$ | 90.02 | 89.23 | 79.66 |

$Y_{2024} = 14{,}845.102 + 248.890X_1 - 251.469X_1^2.$

Similarly, the equation for the effect of nitrogen fertilizer on maize yield:

$Y_{2022} = 15{,}115.402 + 507.927X_2 - 287.767X_2^2$
$Y_{2023} = 15{,}173.730 + 275.302X_2 - 368.183X_2^2$
$Y_{2024} = 14{,}845.102 + 478.774X_2 - 586.648X_2^2.$

Similarly, the equation for the effect of density on maize yield:

$Y_{2022} = 15{,}115.402 + 1{,}660.474X_3 - 1{,}013.066X_3^2$
$Y_{2023} = 15{,}173.730 + 1{,}586.006X_3 - 933.130X_3^2$
$Y_{2024} = 14{,}845.102 + 1{,}379.610X_3 - 1{,}028.403X_3^2.$

From the equation, a one-factor effect plot was obtained (Fig. 2). Different factors and levels had different effects on maize yield and, also, varied among years. In 2022 and 2023, maize yield increased with increasing organic fertilizer content. In 2024, maize yield showed an increasing and then decreasing trend with increasing organic fertilizer content, and maize yield in 2024 was significantly lower than the other two years. This may be related to the rainfall in that year, where sufficient moisture effectively dissolved the nutrients of organic fertilizer in the soil, which in turn contributed to the increase in yield. The maize yield showed a trend of decreasing with increasing nitrogen fertilizer content and planting density. The highest maize yield was recorded at 225 kg hm$^{-2}$ of N fertilizer in 2022, while the highest maize yield was recorded at 150 kg hm$^{-2}$ of N fertilizer in 2023 and 2024. All planting densities were highest at 82,500 plants hm$^{-2}$. It indicates that the appropriate nitrogen fertilizer and density can help to increase maize yield, while too high nitrogen fertilizer application and planting density will rather inhibit the formation of yield.

## Analysis of intercropping effects

The results of ANOVA showed that significant intercropping existed among the crosses. Organic fertilizer-nitrogen fertilizer, organic fertilizer-density and nitrogen fertilizer-density intercropping all affected maize yield (Table 4). The equations for the intercropping effects of organic fertilizer, nitrogen fertilizer, density and yield were obtained using the downscaling method.

Equation for the effect of organic fertilizer-nitrogen fertilizer interactions on yield:

$Y_{2022} = 15{,}115.402 + 450.621X_1 + 507.927X_2 + 127.222X_1^2 - 287.767X_2^2 - 147.271X_1X_2$
$Y_{2023} = 15{,}173.730 + 294.886X_1 + 275.302X_2 - 34.898X_1^2 - 368.183X_2^2 - 10.690X_1X_2$
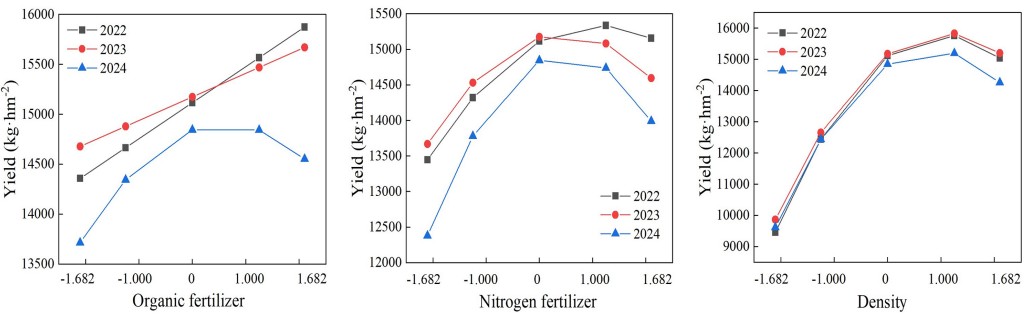

**Figure 2  Single factor effects.**

$$Y_{2024} = 14{,}845.102 + 248.890X_1 + 478.774X_2 - 251.469X_1{}^2 - 586.648X_2{}^2 - 179.584X_1X_2.$$

Equation for the effect of organic fertilizer-density interactions on yield:

$$Y_{2022} = 15{,}115.402 + 450.621X_1 + 1{,}660.474X_3 + 127.222X_1{}^2 - 1{,}013.066X_3{}^2 + 109.674X_1X_3$$

$$Y_{2023} = 15{,}173.730 + 294.886X_1 + 1{,}586.006X_3 - 34.898X_1{}^2 - 933.130X_3{}^2 + 62.085X_1X_3$$

$$Y_{2024} = 14{,}845.102 + 248.890X_1 + 1{,}379.610X_3 - 251.469X_1{}^2 - 1{,}028.403X_3{}^2 - 56.099X_1X_3.$$

Equations for the effect of nitrogen fertilizer-density interactions on yield:

$$Y_{2022} = 15{,}115.402 + 507.927X_2 + 1{,}660.474X_3 - 287.767X_2{}^2 - 1{,}013.066X_3{}^2 + 167.111X_2X_3$$

$$Y_{2023} = 15{,}173.730 + 275.302X_2 + 1{,}586.006X_3 - 368.183X_2{}^2 - 933.130X_3{}^2 + 71.170X_2X_3$$

$$Y_{2024} = 14{,}845.102 + 478.774X_2 + 1{,}379.610X_3 - 586.648X_2{}^2 - 1{,}028.403X_3{}^2 + 194.096X_2X_3.$$

The organic fertilizer-N fertilizer interaction equation yielded a plot of the interaction effect (Fig. 3). The highest yield of maize was obtained when organic and nitrogen fertilizers were at higher levels. In all three years, when the organic fertilizer content was at a high level, the yield showed a trend of increasing and then decreasing with increasing nitrogen fertilizer content. In 2022 and 2023, yield increased with increase in organic fertilizer when nitrogen fertilizer content was higher, whereas in 2024, yield showed an increasing and then decreasing trend with increase in organic fertilizer. It showed that the interaction of organic and nitrogen fertilizer had a synergistic effect on yield at both organic and nitrogen fertilizer levels.

The organic fertilizer-density interaction equation yielded a plot of the interaction effect (Fig. 4). The best yield was obtained only when both organic fertilizer and density were at high levels. At low planting densities, the effect of increasing organic fertilizer content on yield improvement was not significant, indicating that planting density had a greater effect on maize yield. Meanwhile, in all the three years, when the organic fertilizer content was constant, the yield showed an increasing and then decreasing trend with increasing planting density. In 2022 and 2023, when planting density was constant, yield increased
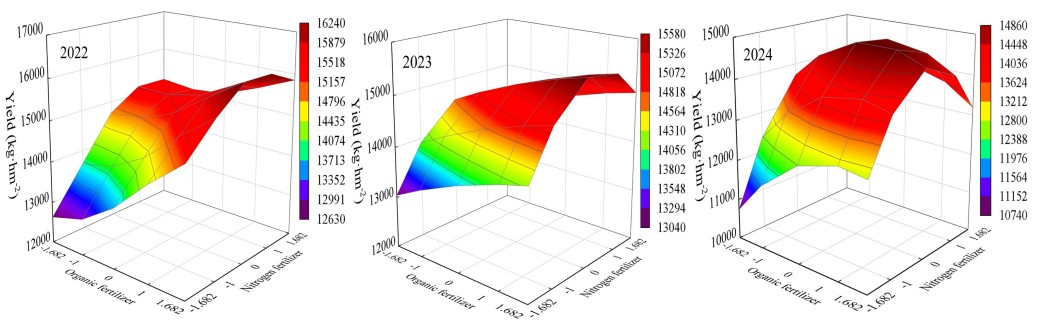

**Figure 3  Diagram of the interaction effect between organic fertilizer and nitrogen fertilizer.**

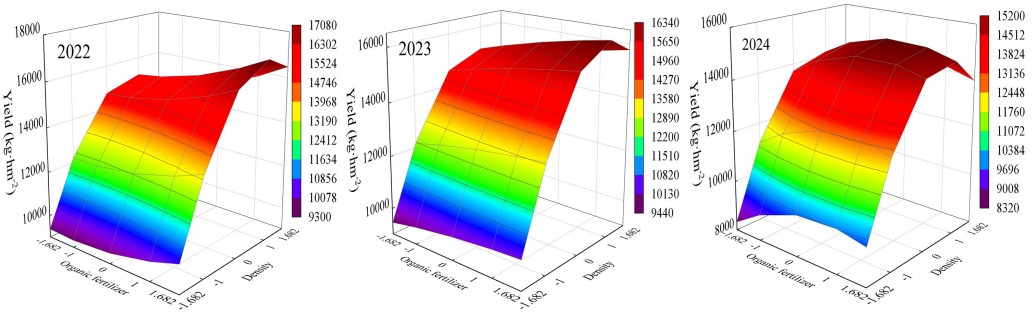

**Figure 4  Diagram of the interaction effect between organic fertilizer and density.**

with increase in organic fertilizer, while in 2024, yield showed an increasing and then decreasing trend with increase in organic fertilizer.

Nitrogen fertilizer-density reciprocal equations yielded a plot of reciprocal effects (Fig. 5). The highest maize yield occurred at higher levels of nitrogen fertilizer and density, indicating a synergistic effect of nitrogen fertilizer and density on yield. In the three years, when the planting density was constant, the yield showed a trend of increasing and then decreasing with the increase of nitrogen fertilizer content. Meanwhile, when nitrogen fertilizer application was constant, yield also showed a trend of first increase and then decrease with increasing planting density. This indicates that both excessive nitrogen fertilizer application and planting density are not conducive to the formation of maize yield.

## Analysis of the optimisation of cultivation measures

With maize yield as a function of the target, the integrated agronomic measures can be optimised by the statistical frequency method to derive the target. Considering the production conditions, climatic conditions, and soil conditions in the Loess Plateau, a maize yield of 15,000 kg hm$^{-2}$ can be positioned as a high-yield target. A total of 24 treatments were calculated to have theoretical yield higher than 15,000 kg hm$^{-2}$ in the experiment, and the frequency of occurrence of the 24 treatments at the level of
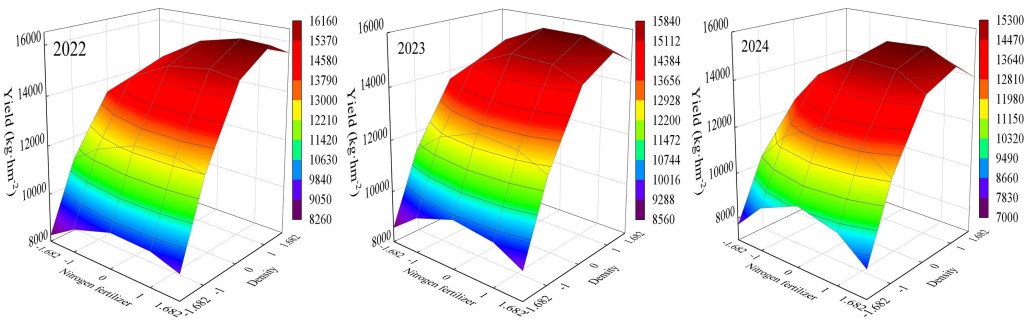

**Figure 5** Diagram of the interaction effect between nitrogen fertilizer and density.

each factor was counted. It can be seen that the highest frequency of yield higher than 15,000 kg hm$^{-2}$ occurred when the coded levels of organic fertilizer, nitrogen fertilizer and density were zero. By calculating the distribution of 95% confidence intervals, stable organic and nitrogen fertilizer application rates and planting densities could be optimised, *i.e.,* organic fertilizer application rates of 6,429.30–7,895.10 kg hm$^{-2}$, nitrogen fertilizer application rates of 159.30–203.55 kg hm$^{-2}$, and planting densities of 72,465–80,940 plants hm$^{-2}$ resulted in high maize yield (Table 6).

## DISCUSSION

### Importance of density on yield

The Loess Plateau, as one of the major areas of dryland food production in China, provides an important guarantee for national food security. Maize is the main food crop in the region, and its yield is the result of a synergy of planting density, nutrient management and environmental conditions (*Jia et al., 2018*). Planting density directly affects population structure and resource use efficiency and is the most important factor affecting yield (*Hou et al., 2020*). This was also confirmed in this study, with the effect on maize yield: density > nitrogen fertilizer > organic fertilizer. Increasing planting density is one of the most direct and effective cultivation measures to improve crop yield. Moderate densification can improve the growth rate and dry matter accumulation of maize in the pre-reproductive period by improving the canopy light interception rate and productivity, but too high a density is not conducive to the accumulation of dry matter in the post-reproductive period, which is prone to cause a decline in the growth rate of the population. At the same time, over-densification will increase the competition among plants for light, water and nutrients, leading to individual stunting and increased risk of failure, which will ultimately affect the formation of yield (*Han et al., 2015*; *Liu et al., 2017*; *Tokatlidis & Koutroubas, 2004*). In this study, maize yield increased with increasing density when planted below 82,500 plants hm$^{-2}$. However, when the density exceeded 82,500 plants hm$^{-2}$, the yield decreased instead with the increase of planting density. Therefore, while increasing planting density, soil fertility and hydrothermal conditions, *etc.*, need to be considered.

**Table 6  Analysis of the optimisation of cultivation measures for high maize yields.**

| Level | Organic fertilizer | | Nitrogen fertilizer | | Density | |
|---|---|---|---|---|---|---|
| | Times | Frequency | Times | Frequency | Times | Frequency |
| −1.682 | 0 | 0 | 0 | 0 | 0 | 0 |
| −1 | 1 | 0.04 | 1 | 0.04 | 0 | 0 |
| 0 | 13 | 0.54 | 14 | 0.58 | 12 | 0.50 |
| 1 | 5 | 0.21 | 6 | 0.25 | 8 | 0.33 |
| 1.682 | 5 | 0.21 | 3 | 0.13 | 4 | 0.17 |
| Sum | 24 | 1 | 24 | 1 | 24 | 1 |
| 95% frequency distribution | 0.191–0.843 | | 0.123–0.714 | | 0.331–0.896 | |
| Optimal range | 6,429.30–7,895.10 | | 159.30–203.55 | | 72,465–80,940 | |

## Effect of water-fertilizer synergy on yield

Water and nutrients are the main influences on crop development and yield formation in arid regions. Insufficient precipitation inhibits nutrient uptake and insufficient nutrients reduce water use efficiency, both of which are detrimental to crop yield formation. (*Mingde, 2019*; *Pinheiro & Chaves, 2011*). In this study, maize yields were significantly higher in 2022 and 2023, when rainfall was more abundant, than in 2024. Under the organic fertilizer treatment, maize yield tended to increase as the amount of organic fertilizer applied increased. This is due to the fact that organic fertilizer addition improves soil structure, enhances water and fertilizer retention capacity and slow release of nutrients, which contributes to kernel filling and resilience in the later stages of maize fertility (*Gathorne-Hardy, 2016*; *Haque et al., 2015*). And with the increase of nitrogen fertilizer application, the maize yield showed a trend of first increase and then decrease. This may be due to the fact that nitrogen addition can quickly replenish key nutrients for crop growth and promote the accumulation of dry matter in maize. Whereas, excessive nitrogen application can cause problems such as vigorous nutrient growth, soil acidification and organic matter loss, which ultimately lead to a reduction in maize yield (*Marques et al., 2017*; *Okalebo et al., 1999*).

## Effect of organic fertilizer, nitrogen fertilizer and density interactions on yield

Reasonable fertilizer application and planting density are key to improving crop yields. Density and fertilizer application interactively influence root spatial distribution. Under high nitrogen conditions, roots tend to concentrate in shallow soil layers, whereas organic fertilizer promotes deep root development, effectively mitigating water competition (*Ma & Zheng, 2018*). Fertilizer supply at high planting densities can effectively alleviate problems such as unbalanced nutrient supply to maize and fertilizer removal at the later stages of fertility (*Lai et al., 2022*). At the same time, organic fertilizers increase the abundance of ammonia-oxidising bacteria by providing a carbon source to stimulate microbial activity, promote organic nitrogen mineralisation, and form a slow-release complementary effect with chemical nitrogen fertilizers to improve nitrogen deficiency under high density conditions. The combination of organic fertilizer and nitrogen fertilizer

can increase bioproductivity and reduce leaching losses, playing an important role in achieving high-yield and green agriculture (*Li et al., 2024*). This study showed that organic fertilizer-nitrogen fertilizer intercropping, organic fertilizer-density intercropping and nitrogen fertilizer-density intercropping all had significant yield enhancement in maize. It is noteworthy that the interaction effect of the three yield increases was significant. For example, at medium densities, organic- nitrogen fertilizers are most effective in increasing yields because of a well-structured population and a balanced supply of nutrients. In contrast, high-density planting may result in lower yields if it is not supported by adequate amounts of fertilizer (*Workayehu, 2000*). Combined with the actual environmental conditions and production conditions in the Loess Plateau, the optimal range of each cultivation measure was obtained when the yield was higher than 15,000 kg hm$^{-2}$: the application of organic fertilizer was 6,429.30–7,895.10 kg hm$^{-2}$, nitrogen fertilizer was 159.30–203.55 kg hm$^{-2}$, and the planting density was 72,465–80,940 plants hm$^{-2}$. Considering the arid climate characteristics of the Loess Plateau, the combination of moderate planting density + slow-release nitrogen fertilizer + organic fertilizer can be adopted, along with appropriate moisture-conserving and rainfall-harvesting cultivation measures (such as double ridge-furrow mulching, straw mulching, *etc.*), to achieve high maize yields. The results are reliable and can provide theoretical basis and practical reference for maize production in the Loess Plateau.

## CONCLUSION

Maize is not only the core crop of the agricultural economy of the Loess Plateau, but also plays an irreplaceable role in ecological protection and livelihood security. The magnitude of contribution of different factors in increasing maize yield was in the following order: density > nitrogen fertilizer > organic fertilizer. Meanwhile, this study screened out the dense fertilizer levels for high yield of maize in the Loess Plateau: 6,429.30–7,895.10 kg hm$^{-2}$ of organic fertilizer, 159.30–203.55 kg hm$^{-2}$ of nitrogen fertilizer, and planting densities of 72,465–80,940 plants hm$^{-2}$. Considering the arid climate characteristics of the Loess Plateau, the combination of moderate planting density + slow-release nitrogen fertilizer + organic fertilizer can be adopted, along with appropriate moisture-conserving and rainfall-harvesting cultivation measures (such as double ridge-furrow mulching, straw mulching, *etc.*), to achieve high maize yields. This study provides theoretical basis and practical experience for the standardised cultivation of maize in the Loess Plateau.

## ACKNOWLEDGEMENTS

The authors thank the experts for editing our article and the anonymous reviewers for their critical comments and suggestions to improve this article.

### Funding
This research was supported by National Key R&D Programme (2023YFD1900403), National Maize Industry Technology System (CARS-02-79), Gansu Provincial Key R&D Programme (24YFNA014), National Natural Science Foundation of China (32360548), and Central Guidance for Local Scientific and Technological Development funding projects (23ZYQA298). The funders had no role in study design, data collection and analysis, decision to publish, or preparation of the manuscript.

### Grant Disclosures
The following grant information was disclosed by the authors:
National Key R&D Programme: 2023YFD1900403.
National Maize Industry Technology System: CARS-02-79.
Gansu Provincial Key R&D Programme: 24YFNA014.
National Natural Science Foundation of China: 32360548.
Central Guidance for Local Scientific and Technological Development: 23ZYQA298.

### Competing Interests
The authors declare there are no competing interests.

### Author Contributions

- Wenbo Mi conceived and designed the experiments, prepared figures and/or tables, and approved the final draft.
- Gang Zhao conceived and designed the experiments, prepared figures and/or tables, authored or reviewed drafts of the article, and approved the final draft.
- Jianjun Zhang performed the experiments, authored or reviewed drafts of the article, and approved the final draft.
- Yi Dang performed the experiments, authored or reviewed drafts of the article, and approved the final draft.
- Lei Wang performed the experiments, prepared figures and/or tables, and approved the final draft.
- Gang Zhou analyzed the data, prepared figures and/or tables, and approved the final draft.
- Shuying Wang analyzed the data, prepared figures and/or tables, authored or reviewed drafts of the article, and approved the final draft.
- Shangzhong Li conceived and designed the experiments, authored or reviewed drafts of the article, and approved the final draft.
- Tinglu Fan conceived and designed the experiments, authored or reviewed drafts of the article, and approved the final draft.
- Jingyu Hu analyzed the data, authored or reviewed drafts of the article, and approved the final draft.
- Xujiao Zhou analyzed the data, authored or reviewed drafts of the article, and approved the final draft.

## Data Availability

The original data are available in the Supplementary File.

## Supplemental Information

Supplemental information for this article can be found online at http://dx.doi.org/10.7717/peerj.19654#supplemental-information.

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
