# Peer review of "Optimal cultivation measures for maize production in the drylands of the Loess Plateau"

_PeerJ, doi:10.7717/peerj.19654_

## Round 0.1 · original submission · Major Revisions

Please revise this manuscript according to the comments raised and re
submit.

Reviewer 1 ·

Basic reporting

1. Temperature is a critical determinant of maize yield; therefore, we recommend including the monthly average temperatures across study years in Figure 1.

2. A brief agronomic profile of the maize variety Xianyu 1483 should be provided, including key parameters such as thousand-seed weight, growth period, and conventional sowing rate.

3. Density and fertiliser application are critical for crop yield formation. I suggest that the authors carefully argue the underlying ecological mechanisms.

4. Figure 2 demonstrates an apparent yield decline in 2024 despite increased organic fertilizer application. This phenomenon may require further explanation.

5. The term "maize" should be consistently used throughout the manuscript, replacing inconsistent references to "corn" (e.g., Lines 31, 32, 102).

6. The Latin binomial nomenclature Zea mays L. should be included at first mention in both the abstract and main text (Introduction section).

7. In Table 6, the column header should read "Optimal range" to maintain formatting consistency with other tables.

8. Harmonize the format of writing numbers throughout the manuscript (e.g., lines 22, 107, 248).

Experimental design

-

Validity of the findings

-

Reviewer 2 ·

Basic reporting

This study examines the impact of nutrient management, planting density, and cultural practices on maize yields and economic efficiency. It uses dry-farmed maize from the Loess Plateau and a three-factor, five-level rotational orthogonal quadratic combination design with organic fertilizer, nitrogen fertilizer, and density. The results show that the contribution of each factor to maize yield is in the order of density > inorganic fertilizer > organic fertilizer. The study explores the effects of different ratios and interactions among organic fertilizer, nitrogen fertilizer, and density on maize yields, providing a theoretical basis and practical experience for standardized cultivation on the Loess Plateau.

1. Introduction
China, the world's second-largest corn producer, plays a crucial role in the global corn industry, particularly in the Loess Plateau, which accounts for 40% of the country's dryland area. Corn cultivation in the region ensures food security but also influences water distribution and the construction of the ecological environment. However, factors such as geography, soil type, and climatic conditions limit corn production.

Low soil fertility is a key factor limiting crop productivity on the Loess Plateau. Fertilizer application, especially nitrogen fertilizer, plays an important role in corn yield. However, excessive nitrogen fertilizer application not only fails to increase corn yield but also causes soil nutrient imbalance and secondary salinization. To address this problem, organic fertilizers have been widely used, which can improve soil fertility, maintain nutrient balance, and reduce environmental pollution.

Selecting fertilizer rates and planting density is critical for improving maize yield on the Loess Plateau. Planting density that is too low does not allow for efficient use of water, fertilizer, light, and heat, and excessive plant spacing can lead to soil erosion. Excessive planting density can destabilize maize production by disrupting soil moisture balance in arid ecosystems and increasing the risk of plant diseases and pests.

This study focuses on dryland maize on the Loess Plateau and examines the effects of fertilizer application and planting density on maize yield, as well as their interactions.


2. Materials and methods

The experiment was conducted from April 2022 to October 2024 at the National Soil Quality Observation Experimental Station of the Institute of Dryland Agriculture, Gansu Academy of Agricultural Sciences. The soil was black loess soil, moderately fertile, with an average annual precipitation of 533.4 mm. The experimental material was corn variety Xianyu 1483. The treatments included three factors at five levels: organic fertilizer, nitrogen fertilizer, and planting density. The experiments were conducted under a wide plastic film mulch and according to a rotating orthogonal experimental design. The corn was sown in mid-April and harvested in late September. The yield per unit area was calculated at 14% moisture. The experimental data were organized, statistically analyzed, and subjected to analysis of variance using Microsoft Excel 2010 and SPSS 27.0 statistical analysis software. The graphics were created with Origin 2021 software.

3. Result

The study aimed to determine the impact of organic fertilization, nitrogen fertilization, and density on corn yield. A regression model was established, with the total regression coefficients reaching a highly significant level. The ANOVA table demonstrated the validity of the models and the reasonable and reliable fit of the regression equations. The contribution rate of each factor to yield was calculated from the F value of each treatment. The contribution rate of each factor to corn yield in 2022-2024 was calculated in the following order: density > nitrogen fertilization > organic fertilization. The contribution of density to corn yield was significantly greater than that of fertilization.

A single-factor effect analysis was conducted, revealing that different factors and levels had different effects on corn yield and varied between years. Corn yield increased with increasing organic fertilization content in 2022 and 2023, but decreased with increasing organic fertilization content and planting density. The highest corn yield was recorded at 225 kg·hm-2 of nitrogen fertilizer in 2022, while the highest yield was recorded at 150 kg·hm-2 of nitrogen fertilizer in 2023 and 2024.

The study analyzed the effects of intercropping on maize yield using ANOVA. The results showed significant intercropping between the crosses, with the organic fertilizer-nitrogen fertilizer, organic fertilizer-density, and nitrogen fertilizer-density interactions all affecting yield. The highest yield was obtained when both organic and nitrogen fertilizer applications were high. The interaction between organic fertilizer and nitrogen fertilizer had a synergistic effect on yield, regardless of whether organic and nitrogen fertilizer applications were high. The best yield was only obtained with high organic fertilizer and density applications.

The highest maize yield was observed with high nitrogen fertilizer and density applications, indicating a synergistic effect of these two types of fertilizer on yield. Excessive nitrogen fertilizer application and excessive seeding density do not promote maize yield.

Integrated agronomic measures can be optimized by the statistical frequency method to determine the target. A maize yield of 15,000 kg·hm-2 can be considered a high yield target on the Loess Plateau. The highest frequency of yields above 15,000 kg·hm-2 was observed when the coded levels of organic fertilizer, nitrogen fertilizer, and density were zero. Stable application rates of organic and nitrogen fertilizer, as well as seeding densities, could be optimized, resulting in high maize yield.

4. Discussion

The Loess Plateau, an important food production area in China's arid zone, is crucial to national food security. Maize, the main food crop, is influenced by the synergy between planting density, nutrient management, and environmental conditions. Planting density directly affects population structure and resource use efficiency, with moderate densification improving growth rate and dry matter accumulation in the pre-reproductive period. However, excessively high planting density can lead to a decline in population growth and competition between plants for light, water, and nutrients, resulting in stunted growth and an increased risk of failure.

In arid regions, water and nutrients are the main factors influencing crop development and yield formation. Insufficient rainfall inhibits nutrient uptake and reduces water use efficiency, thereby impairing yield formation. Corn yields were significantly higher in 2022 and 2023, when rainfall was higher than in 2024. Organic fertilization improved soil structure, enhanced water and fertilizer retention capacity, and slowed nutrient release, contributing to grain filling and resilience in later stages of corn fertility.

Reasonable fertilization and seeding rates are essential for yield improvement. The combination of organic fertilization and nitrogen fertilizer had a synergistic effect on yield regulation. The optimal range for each cultural measure was achieved when yield was above 15,000 kg·hm-2.

5. Conclusion
The study identifies dense fertilizer levels for high maize yield on the Loess Plateau, providing a theoretical and practical basis for standardized cultivation, with planting densities ranging from 72,465 to 80,940 plants·hm-2.

Strengths:
Clarity of Objectives: The reader is made fairly well aware of why maize production in the Loess Plateau is significant and how the situation is appropriately created.

Detailed Methodology: The experimental design, i.e., the factors and their levels, is well explained, and thus the method is easily replicable.

Analytical Rigor: The work makes use of rigorous statistical methods (e.g., ANOVA) to buttress the findings, and graphical presentation using Origin is an additional benefit.
Consistency of Terminology: Within the results and abstract, you show "density > organic fertilizer > organic fertilizer" as the order of contribution of factors. The two instances of "organic fertilizer" resemble a typo and must be removed.

Enhancing Readability:

Break longer paragraphs (for example, the introduction) into shorter segments to make them easier to read.

Add subheadings within sections like "Results" to guide readers through various analyses.

Depth of Discussion

Explain more clearly how the findings can be pragmatically utilised by farmers in the area.

Compare findings with previous research evidence to guide the discussion.

Expansion of Conclusion: Briefly repeat some of the main points of takeaway for scientific and practical use to support the conclusion.

Experimental design

Strengths of Your Experimental Design:
Rotational Orthogonal Quadratic Combination Design: It is a robust methodology that allows us to estimate interactions between many factors (organic fertilizer, nitrogen fertilizer, plant density) and find an optimum resource allocation.

Factors and Levels Choice: Three factors and five levels guarantee investigation covers a wide range of conditions, hence learning something general about their effect.

Controlled Environment: The experiment in the National Soil Quality Observation Experimental Station ensures that soil and other environmental conditions are uniform.

Wide Plastic Film Mulch: Addition ensures less water loss and preservation of soil temperature, which is crucial for research in dryland agriculture.

Refinement Suggestions:
Explanation of Levels: Identify the exact five levels (e.g., amount of fertilizers, planting density) in the manuscript.

Replicates and Randomization: Take precise note of the number of replicates per treatment and whether or not the experimental plots were randomized, since this adds credibility to the results.

Information on Data Collection: Specify what other parameters (other than yield) were tracked, e.g., soil moisture or nutrient levels, to provide added richness to the analysis.

Graphical Representation of Design: It would be helpful for the reader to see a table or a design diagram of the experimental setup.

Validity of the findings

Deductive Statistical Analysis: Application of regression equations and ANOVA provides a strong quantitative framework in the analysis of data. Large regression coefficients and the value of model fit indicate that the inference is data-oriented.

Assessment of Contribution Rate: Calculation of F-value for the determination of contribution made by each factor gives a true report of the contribution they have on maize production, reflecting solid statistical thought.

Controlled Environment: Conducting the experiment at a National Soil Quality Observation Station ensures minimal external interference, thus serving the quality of results.

Reproducibility: Ensuring one has a detailed experimental design and method makes it simple for other researchers to repeat the work, thus making the results more reliable.

Consistency Across Years: While data vary year to year (e.g., optimal concentrations of fertilizer), further verification would also include multi-location trials to correct for variation over environments and climates.

Cross-comparison: Cross-comparison with other studies that have been conducted under similar dryland conditions would complement the verification and report similarity (or difference) to more universal agribusiness studies.

Residual Analysis: Determine whether residual assumptions (i.e., homoscedasticity, independence) of statistical models exist to substantiate your ANOVA results.

Your results seem soundly rooted in the data, but outside confirmations and possible shortfalls noted would make them that much stronger.

Additional comments

Improving the Manuscript:

Abstract Extension: While concise, the abstract would be more helpful if it mentioned principal results informally (e.g., maximum planting densities or specific fertilizer rates).

Technical Terms: Don't hesitate to explain technical terms such as "rotational orthogonal quadratic combination design" to unfamiliar readers.

Visualization:
Use charts or graphs to show the interaction effects of planting density, nitrogen fertilizer, and organic fertilizer on maize yield. Graphics are very capable of capturing the attention of readers.

A flowchart of the experimental process would probably make the process easier to comprehend.

Error Analysis: List all the possible sources of error or weaknesses of the method (e.g., rainfall fluctuations or soil variability) and their possible impact on the findings.

Practical Applications: Highlight how your findings can be applied by farmers or agricultural decision-makers to increase maize yields in dryland areas like the Loess Plateau.

Style and Language:
Professional yet clear so that it will attract a broad readership, i.e., researchers, practitioners of agriculture, and decision-makers.

Use a uniform unit of measurement throughout the paper (e.g., "kg·hm-2" for fertilizer rates of application).

Proofread for minor grammatical inaccuracies or redundancy, for example, duplication of order in the factors' contribution.

Reviewer 3 ·

Basic reporting

Abstract: Tillage is not the treatment, but is mentioned in the abstract as research material. Abstract lacks findings supported by values.

Introduction: A Sufficient description has been given. However, the research gap should be more clearly described.

Methodology: More description is needed. How the field experiment was done, how the data were recorded etc.

Results and discussion: Satisfactory.

Conclusion: Very poor. Must be rewritten.

References: Less care was taken.

Experimental design

-

Validity of the findings

-

Additional comments

I have made several comments on the body of the manuscript that should be addressed during the revision.

Annotated reviews are not available for download in order to protect the identity of reviewers who chose to remain anonymous.

---

## Round 0.2 · accepted · Accept

Reviewers have now commented on your manuscript. In the light of their comments, I am pleased to accept this manuscript for publication.

Reviewer 1 ·

Basic reporting

I remember that I recommended the references that the author needed to read, but I didn't see any response about it. I accept this result. I have no further suggestions and would like to recommend acceptance of this manuscript.

Experimental design

0

Validity of the findings

0